# Investigation on the Role of Pd, Pt, Rh in Methane Abatement for Heavy Duty Applications

Moyu Wang [1,2,*], Panayotis Dimopoulos Eggenschwiler [1], Tanja Franken [3,†], Miren Agote-Arán [3], Davide Ferri [3] and Oliver Kröcher [2,3]

[1] Swiss Federal Laboratories for Materials Science and Technology (EMPA), Automotive Powertrain Technologies Laboratory, CH-8600 Dübendorf, Switzerland; panayotis.dimopoulos@empa.ch

[2] École Polytechnique Fédérale de Lausanne (EPFL), Institute of Chemical Sciences and Engineering, CH-1015 Lausanne, Switzerland; oliver.kroecher@psi.ch

[3] Paul Scherrer Institut, Forschungsstrasse 111, CH-5232 Villigen, Switzerland; tanja.franken@psi.ch (T.F.); miren.agote-aran@psi.ch (M.A.-A.); davide.ferri@psi.ch (D.F.)

* Correspondence: moyu.wang@empa.ch

† Present address: Department of Chemical and Biological Engineering, Friedrich-Alexander University Erlangen-Nürnberg (FAU), 91058 Erlangen, Germany.

**Abstract:** Methane abatement remains a challenge in aftertreatment systems of natural gas engines, currently under discussion in combination with synthetic methane. In this study, Pt/Rh and Pd/Rh-based three-way catalysts are investigated under various transient conditions because transients between $O_2$ excess (lean) and $O_2$-poor (rich) conditions can significantly enhance methane abatement. At mid to high temperatures, transitions from rich to lean feed yield higher rates of methane direct oxidation under lean conditions with the Pt/Rh catalyst, compared to the Pd/Rh catalyst. Both catalysts are able to trigger methane steam reforming (SR) after transitions from lean to rich feed. The SR reaction leads to increased $H_2$ and $NH_3$ formation. However, SR deactivates much faster in the Pt/Rh catalyst. At low temperature, the Pt/Rh catalyst is more active for SR. Results from an additional Pd-only catalyst confirm that Rh is essential for $NO_x$ conversion and high $N_2$ selectivity. The distinct characteristics of Pt, Pd and Rh demonstrate the benefits obtained from the combination of the three platinum group metals. The potential of the Pt/Pd/Rh catalyst is proved to be significant throughout the complete engine map. Under optimized lean/rich oscillatory conditions, the Pt/Pd/Rh catalyst yields more than 95% methane conversion under almost all conditions while maintaining efficient abatement of all other pollutants.

**Keywords:** palladium; platinum; rhodium; catalytic methane oxidation; steam reforming; lean/rich oscillations; ammonia formation

## 1. Introduction

In recent years, synthetic methane has been viewed as a potential energy carrier to solve the problem of renewable energy storage [1,2]. Biomethane has also been mentioned frequently as an important piece in the transition to a carbon-neutral society [3]. In combination with the widely discussed obstacles in complete electrification of the heavy-duty sector [4,5], a combustion engine fueled with renewable methane is an attractive solution for its decarbonization. Natural gas, which consists of around 90% methane ($CH_4$), can be viewed as an intermediate fuel. Compared to diesel, natural gas results in lower energy specific $CO_2$ emission [6]. Whether natural gas or renewable methane, the stable tetrahedral structure of the $CH_4$ molecule poses challenges for the aftertreatment system. Oxidation of $CH_4$ is more difficult than longer chain hydrocarbons in the exhaust of conventional diesel or gasoline engines [7].

In view of stringent emission standards, stoichiometric natural gas engines attract more attention than diesel engines; this is due to the simultaneous high $NO_x$ and CO

conversion obtained with three-way catalysts (TWC). Under stoichiometric conditions, ammonia formation in the catalyst requires additional consideration [8,9] because this compound is strictly regulated in EURO 6 emission standards [10].

The majority of the state of art stoichiometric natural gas aftertreatment system consists of heavily loaded Pd/Rh catalyst [8,9,11]. Rhodium is well known for its highly efficient selective reduction of $NO_x$ [12,13]. Palladium has been found to be the most active noble metal for methane oxidation [14,15]. It is agreed [16–18] that a mixture of Pd/PdO results in the highest $CH_4$ oxidation activity. A temperature hysteresis of $CH_4$ conversion has been reported [19–21], which was attributed to the different oxidation states of palladium. PdO is found to be unstable above 800 °C and decomposes into metallic Pd(0). The reoxidation of Pd to PdO occurs only below 600 °C [19]. Apart from methane direct oxidation, palladium and rhodium can catalyze methane steam reforming (SR) [22,23]. Ceria-based mixed oxides, which are commonly used in TWCs as support for their oxygen storage capacity, are identified as SR promoters [22,24]. During SR reactions, $CH_4$ is adsorbed on noble metal sites, while $H_2O$ is dissociatively adsorbed on reduced ceria sites [22,25,26]. SR was hinted to play an important role in methane abatement at near stoichiometry regions [27]. Our previous work [28] has found that in a ceria supported Pd/Rh catalyst, SR contributes significantly to $CH_4$ conversion in transient transitions from lean to rich conditions. However, SR attenuates with time due to accumulated carbonaceous species on the ceria surface. Based on the temporary high SR reaction rate, λ oscillations across stoichiometry have been proposed to achieve lasting high $CH_4$ conversion [29]. λ conversion enhancement during λ oscillations was reported in several other works [17,27,30].

Despite the many attempts to understand $CH_4$ oxidation catalysts, a comprehensive understanding of different roles played by Pt, Pd, and Rh during $CH_4$ conversion is lacking, in particular for the temperature range relevant to natural gas aftertreatment. In this work, various catalysts with different platinum group metal compositions were tested under various feed conditions and at different temperatures. Characteristic behaviors of Pt, Pd and Rh were identified. Based on the identified characteristics, an improved catalyst composition was analyzed under optimized λ oscillation strategies during a world harmonized steady cycle (WHSC) test.

## 2. Methods

### 2.1. Measurements on Engine Test Bench

#### 2.1.1. Catalysts

Catalysts used in this study are commercial honeycomb catalysts containing different combinations of platinum group metals supported on alumina and ceria zirconia: Pd-only (62 g/ft$^3$), Pt/Rh (44/4.5 g/ft$^3$), Pd/Rh (64.4/4.6 g/ft$^3$), and Pt/Pd/Rh (22/32.2/4.5 g/ft$^3$). All catalysts are wash coated on ceramic monolith substrates (400 cpsi). The diameter of the substrate is 105 mm. The monolith lengths are either 50 mm or 100 mm, depending on the setup. The cross sections of the catalysts were studied with SEM (Hitachi TM3030Plus).

#### 2.1.2. Engine Test Bench Setup

Catalysts are investigated downstream of the turbocharger of an OEM stoichiometric natural gas engine. The engine has a bore of 95.8 mm. The total displacement is 2998 cm$^3$. Natural gas from the grid is used, whose contents are frequently controlled by gas chromatography. $CH_4$ is the largest component (91 vol%), followed by C2 hydrocarbons (4.7 vol%) and C3 hydrocarbons (1.1 vol%).

Gas sampling lines are installed at both the inlet and the outlet of the catalysts. Sampled gases are simultaneously analyzed by three instruments: a Horiba-MEXA (combined system of nondispersive infrared sensor, chemiluminescence detector, and paramagnetic oxygen analyzer and flame ionization detector), a AVL FTIR and a Hsense mass spectrometer. All measurements are read out at a frequency of 10Hz. The measurement delays of each species in each instrument are determined by targeted tests and are corrected

in transient concentration measurements. A λ sensor (Bosch LSU 4.9) is located at the inlet of the catalyst, which measures instantaneous lambda values ($\lambda_{sensor}$) and is used to control engine operations. If not further specified, the λ values given in the following text correspond to λ sensor signals. Depending on temperature, pressure and exhaust gas compositions, the λ sensor signal has inevitable small deviations (<1%) to real λ values [31]. Temperature at inlet and outlet of the catalyst are measured by thermocouples.

### 2.1.3. λ Ramp Measurements

The measurement procedure for λ ramps starts with a pre-conditioning period of 10 min, in which the engine runs at steady state at λ = 0.9 for 10 min. This guarantees that the λ ramp starts with known catalyst state and fully depleted oxygen storage capacity (OSC). After preconditioning, the λ value of the exhaust gas increases linearly from 0.9 until 1.1 in 15 min, while keeping the engine speed and torque steady. Upon reaching λ = 1.1, the λ value starts decreasing linearly toward 0.9 and reaches 0.9 in 15 min. The transient concentration profiles are recorded during the λ sweeps.

### 2.1.4. Cold Start Measurements

Cold starts are performed after the test bench is cooled overnight. During cold starts, the engine operation point is kept at 1600 rpm, 50 Nm, and the λ sensor signal stays at 1.00. The transient concentration profiles are recorded until 30 min after the start of fuel injection.

### 2.1.5. Slow/Fast λ Oscillations

During λ oscillations, λ sensor signal makes step changes between $\lambda_c \pm 0.025$ ($\lambda_c$ is the pre-determined λ center). The duration of lean/rich periods are kept the same. The lean/rich duration varies from less than 3 s to 60 s, according to measurement needs. Before performing each λ oscillations measurement, the catalyst is pre-conditioned at λ = 0.90 for 10 min. After the start of λ oscillations, the concentration profiles are monitored until reaching stabilization. All analyses are based on stabilized concentration profiles.

### 2.2. *Temperature Ramp Measurements in Model Gas Reactor*

A cylinder piece (1.39 cm$^2$ entrance area, 4.83 cm length) was extracted from the Pd/Rh monolith mentioned in engine test bench tests. The piece was then tested in a model gas reactor which comprises a gas mixing unit connected to a reactor allowing for individual adjustments of $H_2$ (diluted in $N_2$), CO, NO, $O_2$, $CH_4$ and $N_2$ concentrations. Outlet concentrations are analysed using a mass spectrometer (Pfeiffer, Omnistar; Hesse, Germany) and a FTIR spectrometer (iS50, Thermo; Waltham, MA, USA). The setup has been described in detail in our previous work [28]. The catalyst was analyzed under a slightly rich gas mixture (1500 ppm $CH_4$; 9000 ppm CO; 2750 ppm NO; 5000 ppm $H_2$; 7600 ppm $O_2$; 50,000 ppm $H_2O$, bal. $N_2$) in temperature sweep experiments (100 °C → 400 °C; 5 °C/min). The calculated λ value was 0.995. The flow rates of gases were adjusted to obtain a gas hour space velocity (GHSV) of 75,000 h$^{-1}$. The GHSV is defined as the ratio of feed gas volume flow rate (derived from the mass flow rate and the measured temperature) to the catalyst volume.

## 3. Results and Discussions

### 3.1. *SEM Washcoat Characterization*

Figure 1 shows the SEM image of the Pd/Rh catalyst, in which the substrate and washcoat can be clearly distinguished. The SEM analysis revealed that the washcoat thickness in the channel corners is significantly higher than in the central parts. In the corner of the cells, the washcoat accumulates a thickness of up to 250 μm; in contrast, the layer thickness in the center of the substrate walls is below 80 μm. All investigated cells show comparable washcoat distribution and morphology.

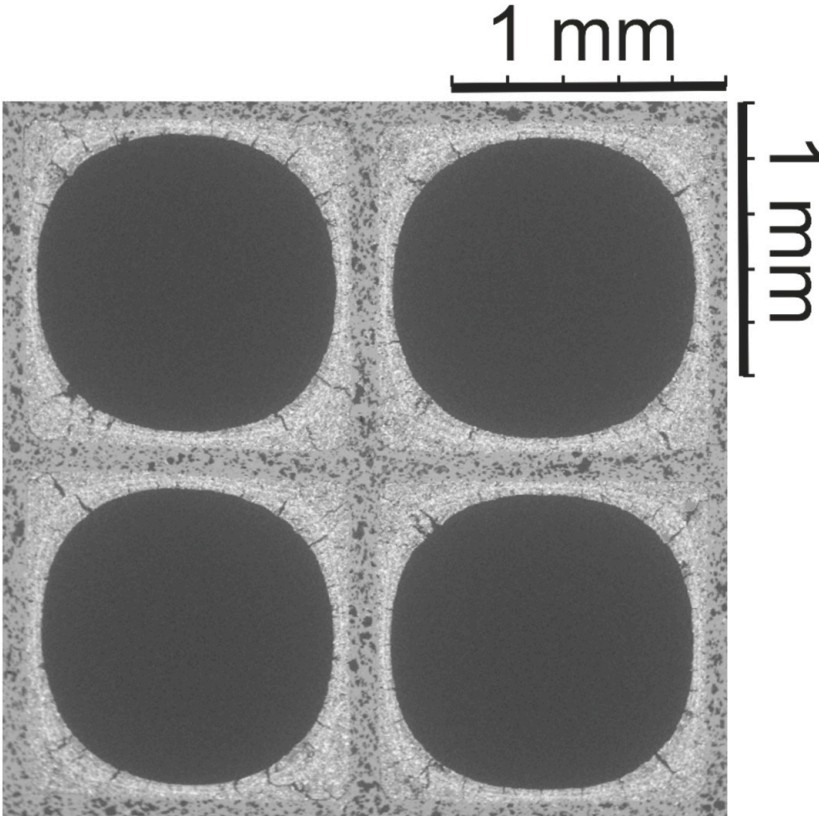

**Figure 1.** SEM image ($\times$30) of the Pd/Rh catalyst.

### 3.2. λ Ramp in the Mid Temperature Range

Figure 2 shows CH$_4$ conversions obtained during λ ramp-up and ramp-down for Pd/Rh (Figure 2a) and Pt/Rh (Figure 2b) catalyst (50 mm length). The engine operates at a speed of 1600 rpm and 210 Nm torque. The catalyst inlet temperature is approximately 520 °C, which varied by $\pm$10 °C according to the λ values. The gas hourly space velocity (GHSV) is of ca. 256,000 h$^{-1}$. The inlet concentrations are listed in Appendix A for λ values of 0.01 intervals. During λ ramp-up in Pd/Rh catalyst (blue line in Figure 2a), CH$_4$ conversion stays 0 until λ is slightly larger than 1.0. This demonstrates that both conversion pathways of CH$_4$ (direct oxidation by oxygen and SR) are inactive. Direct oxidation is restricted by the lack of oxygen under rich conditions, while SR is likely inhibited by surface carbonaceous species [28]. At λ slightly larger than 1.0, a sharp increase of CH$_4$ conversion is observed. Similar observations have been made in our previous studies [28] during a quasi-steady state λ sweep on the same Pd/Rh catalyst. The sharp rise in conversion was proved to be linked to the oxygen excess remaining after oxidizing other active reducing agents, e.g., CO and H$_2$. Until this point in time, palladium is probably in metallic phase, and its surface is not yet fully covered by oxygen molecules. The sharp rise in CH$_4$ conversion lasts briefly and is followed by a sharp decrease in CH$_4$ conversion. This can be explained by oxygen inhibition, i.e., oxygen molecules cover the active sites and hinder CH$_4$ adsorption [32,33]. With increasing λ value, Pd is slowly oxidized to PdO due to the excess oxygen in lean feed. Between 500 °C and 600 °C, Pd oxidation is possible but is slow [34,35]. It is generally agreed that the combination of metallic Pd(0) and PdO is the most active palladium state for lean CH$_4$ oxidation [16–18]. Therefore, in our λ ramp-up, on the lean side, the CH$_4$ conversion first increases to a maximum of around 0.27, probably corresponding to the existence of a mixed Pd/PdO phase, and then it decreases slowly to around 0.13 at λ = 1.1 when Pd is likely completely oxidized to PdO.

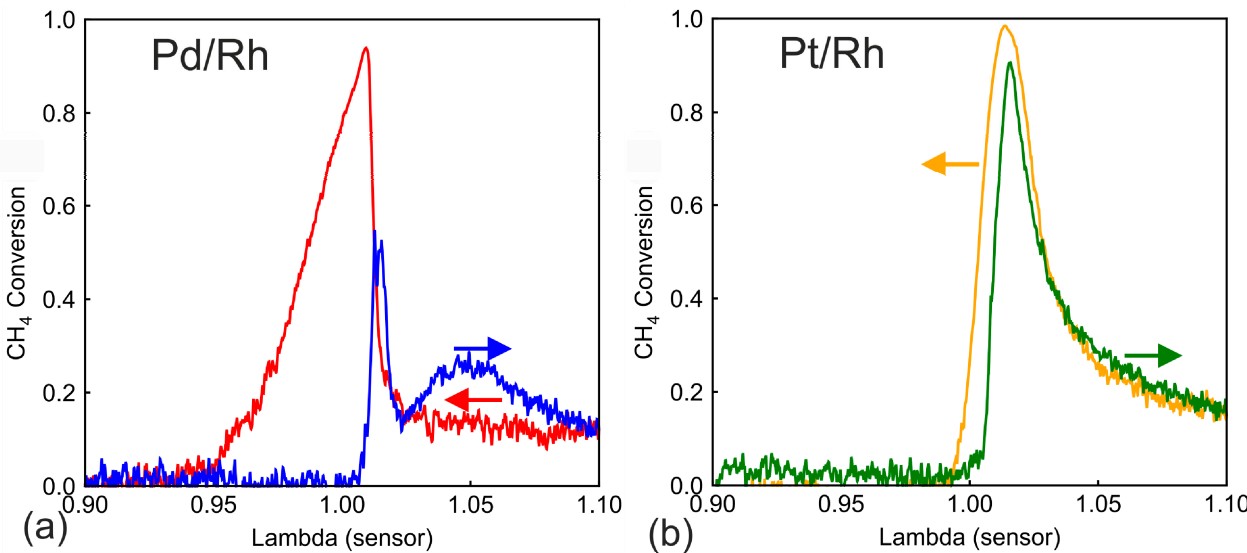

**Figure 2.** $CH_4$ conversion of the (**a**) Pd/Rh (**b**) Pt/Rh catalyst during $\lambda$ ramp-up (blue and green) and ramp-down (red and yellow) at 520 °C.

During $\lambda$ ramp-down, $CH_4$ conversion shows clear hysteresis in the lean phase compared to ramp-up, and it stays steady at around 0.13 conversion. The hysteresis demonstrates different palladium oxidation states between ramp-up and ramp-down. While ramping down, palladium stays fully oxidized in PdO state. Several previous studies have shown that the oxidation of $CH_4$ is a zero-order reaction to oxygen concentration [33,36], which is in line with the steady lean $CH_4$ conversion during ramp-down. At $\lambda$ sensor slightly higher than 1.00, $CH_4$ conversion rises sharply to more than 0.9. The $CH_4$ conversion during ramp-down is significantly higher, compared to zero $CH_4$ conversion between $0.95 < \lambda < 1.00$ during ramp-up. The high $CH_4$ conversion at around $\lambda = 1.00$ only decreases slowly in the rich phase, and it reaches zero conversion at $\lambda < 0.95$. The same strong enhancement of $CH_4$ conversion was studied in detail in our previous work [28]. Activation of SR reaction pathway is proved to be the cause. SR reaction, however, only lasts a short period of time, until all active ceria sites are blocked by surface carbonaceous species (mostly carbonates). The carbonate-blocked ceria sites can be regenerated during lean conditions, which explains the activation of SR only after shifting from lean to rich conditions. The $H_2O$ content varies only slightly, from 17.3 vol% (minimum at $\lambda = 1.1$) to 19.0 vol% (maximum at $\lambda = 1.0$), when $\lambda$ value changes from 0.9 to 1.1. It is safe to neglect the influence of $H_2O$ variation.

During $\lambda$ ramp-up with the Pt/Rh catalyst (Figure 2b green line), similar to Pd/Rh, no $CH_4$ conversion occurs until the sharp onset of conversion due to sufficient $O_2$ at $\lambda$ slightly higher than 1.00. The lean $CH_4$ conversion profiles of the Pt/Rh catalyst are less complicated than Pd/Rh. $CH_4$ conversion decreases steadily until $\lambda = 1.1$. The $CH_4$ conversion in the vicinity of $\lambda = 1.00$ is much higher than that of Pd/Rh. The active state of platinum for $CH_4$ oxidation is metallic Pt [37]. Due to instability of $PtO_x$ above 300 °C, platinum is expected to stay in metallic state during the experiment [15]. Even though oxygen inhibition affects also the Pt/Rh catalyst, Pt is in its in active form after the rich/lean transition and converts $CH_4$ more efficiently than Pd/Rh at near stoichiometric lean regions. During ramp-down, no hysteresis is observed on the lean side, most probably due to the fact that Pt stays in the same metallic form. On the rich side ($\lambda < 1.0$), the $CH_4$ conversion is near zero in the Pt/Rh catalyst for both ramp-up and ramp-down, signaling either low activity toward SR or very fast deactivation of SR.

Conversion of CO and $NO_x$, as well as outlet concentration profiles of $H_2$ and $NH_3$ during ramp-up, are plotted in Figure 3. Unsurprisingly, in both catalysts, CO (Figure 3a) is 100% converted under lean conditions, while the conversion is low at $\lambda < 1.0$. In our

previous work [28], CO was proved to undergo mainly the water gas shift (WGS) reaction under very rich conditions. Pt/Rh exhibits higher activity in WGS, despite having less precious metal loading (48.5 g/ft3) compared to the Pd/Rh catalyst (70 g/ft3). The $NO_x$ conversion profile for both catalysts is very similar, with almost 100% conversion on the rich side and low conversion in the lean side. $H_2$ in the exhaust gas are first oxidized by $O_2/NO_x$ when either is available. In the downstream part of the catalyst, $H_2$ is formed through WGS reaction [28]. Due to slightly better WGS behavior of Pt/Rh, the outlet $H_2$ concentration is also slightly higher. $NH_3$ is not detected in the engine out exhaust and is produced in the catalysts via $NO_x$ reduction [38,39]. The $NH_3$ production profile (Figure 3d) is almost the same for Pt/Rh and Pd/Rh catalysts.

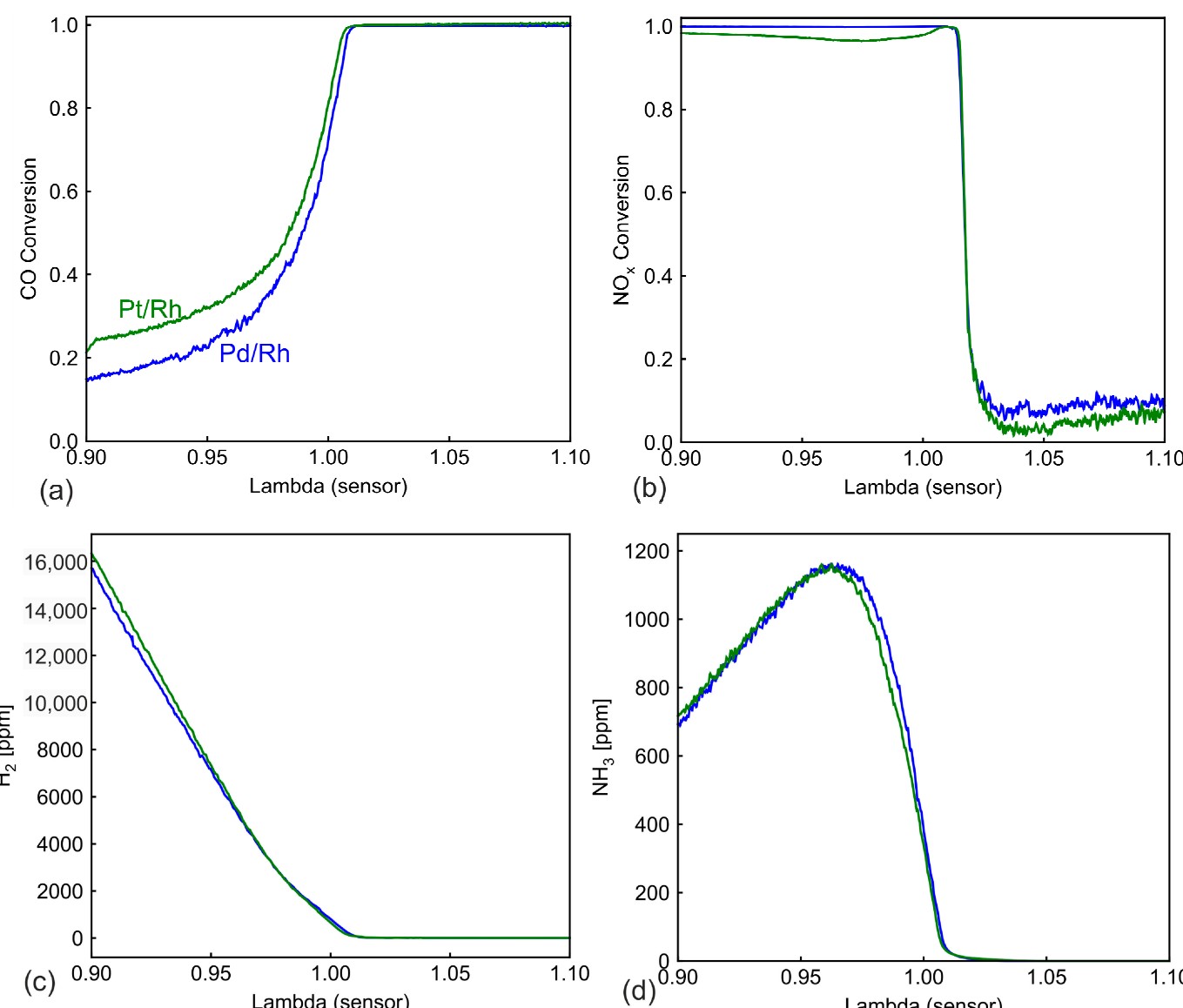

**Figure 3.** Comparison of (**a**) CO conversion (**b**) $NO_x$ conversion in the Pt/Rh and the Pd/Rh catalyst and (**c**) $H_2$ concentration and (**d**) $NH_3$ concentration downstream the catalysts during λ ramp-up at 520 °C.

To better understand the role of Rh in Pd/Rh, the same λ ramp is performed on a Pd-only catalyst. In Figure 4, $CH_4$ conversion profiles demonstrate almost identical characteristics as the Pd/Rh catalyst (Figure 2a). During ramp-up, there is an initial conversion peak at around 1.00, followed by the lowest conversion shortly after the peak.

With increasing oxygen in the feed, the $CH_4$ conversion rises slowly, reaching a maximum value briefly, and subsequently decreases to a steady value. This is additional evidence that the characteristic behavior of $CH_4$ conversion on the lean side is attributed to palladium oxidation states, while presence of Rh has no significant effect.

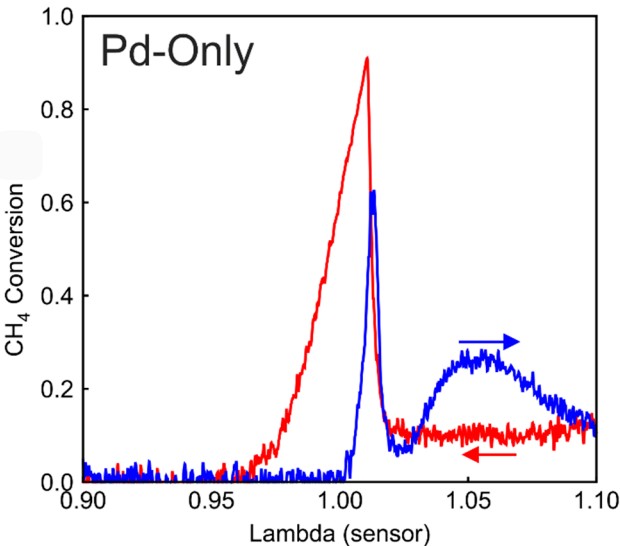

**Figure 4.** $CH_4$ conversion during λ ramp-up (blue) and λ ramp-down (red) in the Pd-only catalyst.

Figure 5 shows a comparison of $NO_x$ conversion and $NH_3$ selectivity between the Pd/Rh and Pd-only catalyst. The Pd-only catalyst exhibits $NO_x$ conversion lower than 60% at λ < 0.96, whereas the $NO_x$ is nearly completely converted with the Pd/Rh catalyst. At very rich conditions (λ < 0.93), $NH_3$ selectivity is nearly 100% for both catalysts. As λ increases toward 1.00, $NH_3$ selectivity of Pd/Rh catalyst decreases much faster than in the Pd-only catalyst. $NH_3$ selectivity of the Pd/Rh catalyst is significantly lower in regions within 3% of stoichiometry, which is the relevant range for real-life operating conditions. The results are in line with the notion that Rh is the most active platinum group metal for selective $NO_x$ reduction to $N_2$ [12,13,40]. In view of strictly regulated $NO_x$ and $NH_3$ emissions in engine exhausts, Rh is an important component alongside Pd and/or Pt.

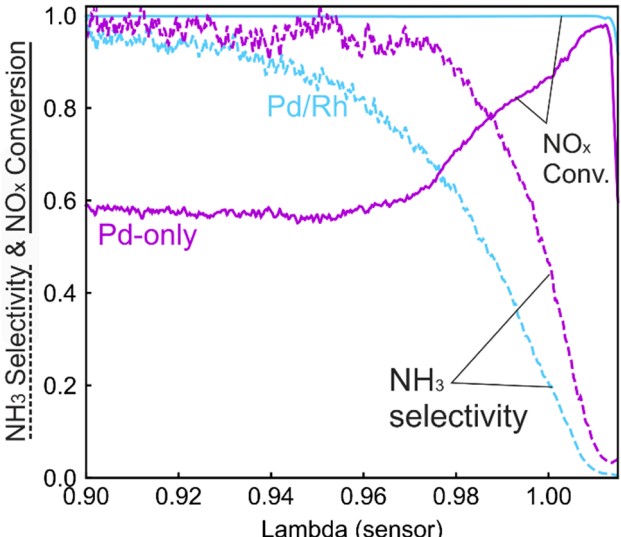

**Figure 5.** Comparison of $NO_x$ conversion and $NH_3$ selectivity during λ ramp-up in the Pd-only and Pd/Rh catalyst.

### 3.3. λ Ramps at Low and High Temperature

The λ ramp procedure was repeated at different engine loads for analyzing the temperature effect. In Figure 6a, the engine operates at high load (2800 rpm 220 Nm), at a catalyst inlet temperature of ca. 630 °C and a GHSV of approximately 546,000 h$^{-1}$. In Figure 6a (high temperature), during λ ramp-up of the Pd/Rh catalyst (blue curve), the CH$_4$ conversion decreases monotonously after the initial conversion peak at λ slightly higher than 1.00, in contrast to the fluctuating lean conversion in Figure 2a. There is also no lean conversion hysteresis. This phenomenon could be attributed to the instability of PdO at high temperatures. The oxidation from Pd to PdO only occurs below 600 °C [19]. The catalyst temperature in Figure 6a lies at 630 °C or higher; therefore, palladium stays in the form of metallic Pd(0), even under lean conditions. The CH$_4$ conversion of the Pd/Rh catalyst at λ = 1.1 in Figure 6a is almost the same as that in Figure 2a, even though the temperature is much higher. This is more evidence of Pd staying in metallic state, as Pd(0) is demonstrated to be less active in CH$_4$ oxidation than PdO [34]. In comparison, at high temperatures (Figure 6a), Pt/Rh catalyst exhibits higher CH$_4$ conversion than in Figure 2a. This is rather unsurprising because metallic Pt is the active site for both temperatures. The SR hysteresis under rich conditions is again observed with the Pd/Rh catalyst, but only in a narrower λ range. SR deactivates faster at higher engine loads due to higher GHSV and temperature, which correspond to results from our previous study [28].

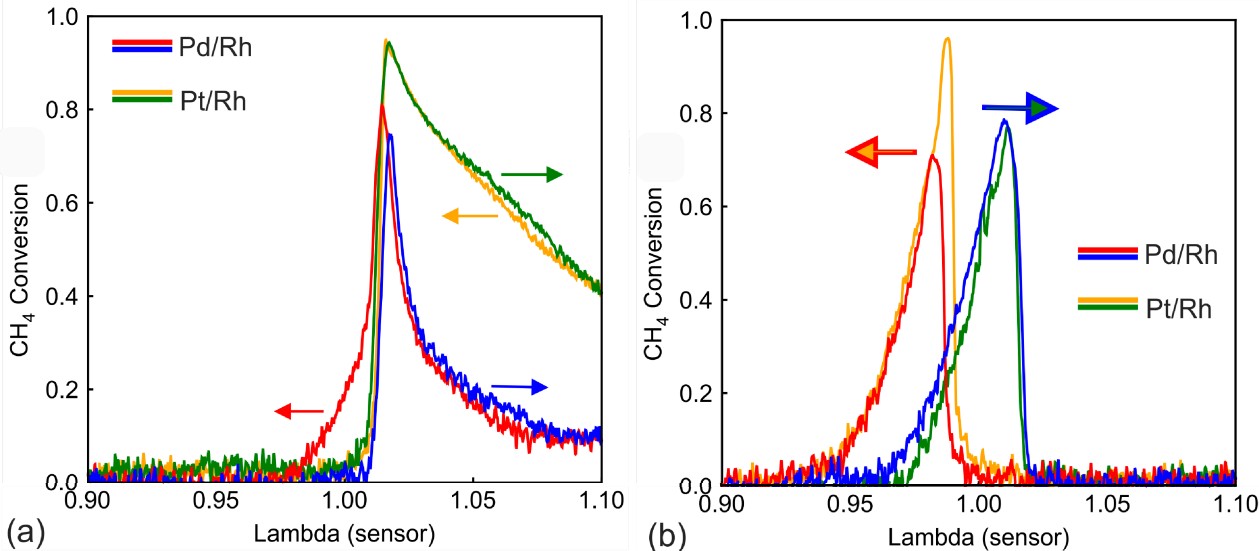

**Figure 6.** CH$_4$ conversion during λ ramp-up and λ ramp-down in the Pd/Rh and Pt/Rh catalyst, at temperature (**a**) 630 °C (**b**) 360 °C.

Figure 6b shows the CH$_4$ conversion during λ ramp under low load conditions (1600 rpm 50 Nm) at an inlet temperature of ca. 360 °C and a GHSV of ca. 83,000 h$^{-1}$. Under lean conditions, CH$_4$ is not converted at all for Pd/Rh and Pt/Rh catalysts, which signals that the temperature is too low for direct oxidation with oxygen. At near stoichiometry and under rich conditions, a strong conversion peak (over 65%) is observed for both ramp directions and catalysts. The Pt/Rh catalyst displays higher conversion during ramp-down. Because the temperature is too low for direct oxidation, these conversion peaks are based on the SR reaction. In contrast to higher temperatures, SR is activated during λ ramp-up. This implies that the proposed deactivation mechanism through carbonaceous formation cannot completely block SR at low temperatures. The exact mechanism requires further studies. In Figure 6b, there is a hysteresis of conversion profiles in the form of displacement between ramp-up and ramp-down. The displacement at the right edge of the conversion can be easily attributed to OSC. However, the conversion discrepancy lasts even after OSC is fully depleted (λ < 0.97). Figure 7 plots CO concentration together with methane conversion for

both the Pd/Rh (Figure 7a) and the Pt/Rh (Figure 7b) catalyst. CO concentration (dotted lines) exhibits a long-lasting hysteresis throughout the entire rich side. The hysteresis can be explained with the difference in WGS reaction rate, caused by the accumulated amount of slow-forming surface carbonate species [41–43]. The hysteresis of $CH_4$ conversion is linked to the outlet CO concentration. This is demonstrated in Figure 7, where two characteristic $CH_4$ conversions (0.1 and 0.2) are selected, and the corresponding CO concentrations are indicated. With the Pd/Rh catalyst (Figure 7a), at 0.1 $CH_4$ conversion, the corresponding CO concentration is about 500 ppm for both ramp-up and ramp-down. The same applies to 0.2 $CH_4$ conversion: CO concentration is about 350 ppm for both ramp-up and ramp-down. Additionally, in Figure 7b (Pt/Rh catalyst), the characteristic $CH_4$ conversion (0.1 or 0.2) corresponds to the same CO concentration for ramp-up and ramp-down (within the same catalyst). This suggests that the low temperature $CH_4$ conversion is linked to the CO concentration, which has a strong inhibition effect on SR [44,45]. At mid to high temperatures (Figures 2 and 6a), due to full deactivation of SR during ramp-up, the effect of CO inhibition is not observed.

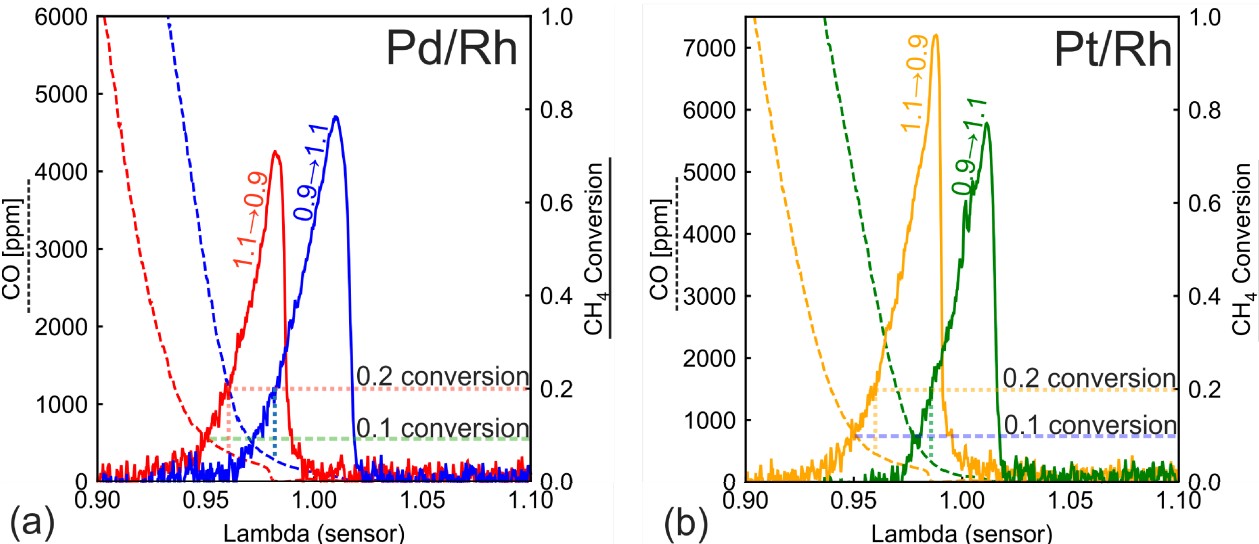

**Figure 7.** CO concentration and $CH_4$ conversion during λ ramp-up and λ ramp-down in (**a**) the Pd/Rh and (**b**) the Pt/Rh catalyst at temperature 360 °C.

### 3.4. Analysis of Low Temperature Behavior Based on Cold Start and Temperature Ramps

In model gas reactors, temperature-programmed techniques are frequently used to analyze reaction mechanisms. Engine test benches are less flexible in changing only one operating parameter because temperature, GHSV and engine out emissions are all interconnected. Cold start tests provide a unique opportunity where temperature increases continuously with stable engine out emissions. In Figure 8, evolution of outlet temperature and species concentrations are plotted against inlet temperature during cold start. In the top plot of Figure 8, the outlet temperatures of the Pd/Rh and the Pt/Rh catalysts (blue and green line, respectively) are depicted together with the inlet temperature (black linear line). At the beginning of the cold start process (inlet temperature < 130 °C), no chemical reactions can take place. Heat is transferred from the hot exhaust gas to the cold catalyst and the catalyst canning. The inlet temperature is higher than the outlet temperature. As the inlet temperature rises, exothermic reactions, e.g., $H_2$ and CO oxidation, start to occur, which results in higher outlet temperatures than the inlet (>175 °C). Detailed investigations and analysis of heat transfer phenomena during cold starts can be found in [46]. For the Pd/Rh catalyst (blue line), $NO_x$ conversion starts at around 130 °C. At the same time, $H_2$ consumption and $N_2O$ production are observed. As CO is not unconverted at this temperature, the observed $NO_x$ is solely reduced by $H_2$ leading to $N_2O$ formation. $N_2O$ is often observed as an intermediate product of $NO_x$ reduction and can be re-adsorbed

and further reduced to either $N_2$ or $NH_3$ [47,48]. At ca. 150 °C, $NH_3$ starts to appear, and its concentration reaches a maximum value at ca. 165 °C before decreasing to zero at ca. 200 °C. The decrease in $NH_3$ can be attributed to two effects: the complete consumption of $H_2$ by $O_2$ ($H_2$ concentration plot) and the oxidation of $NH_3$ by $O_2$. Our previous study on the same Pd/Rh catalyst suggests that the oxidation of $NH_3$ starts above 150 °C and below 200 °C [49]. CO starts to be consumed at approximately 160 °C, much later than the start of the consumption of $NO_x$, $O_2$, and $H_2$.

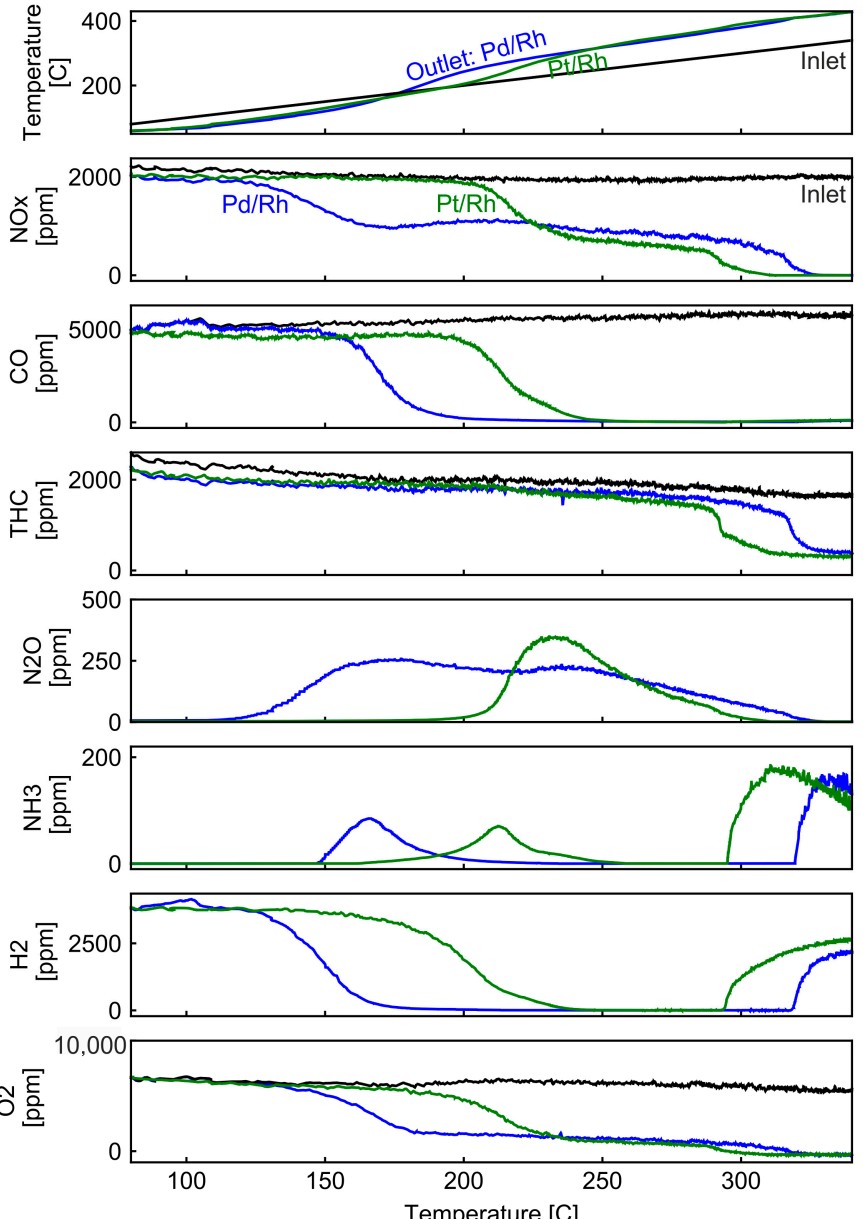

**Figure 8.** Comparison of temperature and concentration profiles upstream and downstream the Pd/Rh and Pt/Rh catalysts during cold start experiment.

In the Pt/Rh catalyst, the start of conversion for $NO_x$, CO and $H_2$ occurs at approximately 50 °C higher temperature, in comparison to the Pd/Rh catalyst. The different platinum group metal loading of the two catalysts makes it difficult to compare their oxidation activity. With the Pt/Rh catalyst, $NO_x$ consumption occurs at a similar temperature as CO, and much later than the start of $O_2$ consumption. $N_2O$ also appears at a much higher temperature (80 °C) than in the Pt/Rh catalyst. First principle calculation has shown that $N_2O$ binds more strongly to Pt than Pd and desorption is more difficult with Pt [48].

THC conversion starts at around 290 °C in the Pt/Rh catalyst, around 30 °C lower than the Pd/Rh catalyst. The start of conversion of total hydrocarbons (THC, more than 90% methane) accompanies immediate formation of $NH_3$ and $H_2$, suggesting that THC are consumed by SR, rather than by oxidation with oxygen. The results agree with the above conclusions indicating that the only reaction pathway for $CH_4$ is SR during $\lambda$ ramps at 360 °C. In view of the fact that Pt/Rh catalyst has lower metal loading but lower SR initiation temperature, it is safe to conclude that Pt has better SR activity at low temperatures than Pd.

The Pd/Rh catalyst is further studied in the model gas reactor with a controlled temperature ramp (5 °C/min). In Figure 9, the results of concentrations and conversion of major pollutants are shown. Similar to the Pd/Rh catalyst during cold start, $N_2O$ is the first product observed at low temperature, before $N_2$ and $NH_3$ is formed. This confirms that $N_2O$ is an intermediate product of $NO_x$ reduction. At low temperature (<200 °C), $NH_3$ concentration peak at around 168 °C and decreases quickly to zero with rising temperature, in agreement with the results obtained on the engine test bench. At 168 °C, the sudden decrease in $NH_3$ concentration is accompanied by a simultaneous increase in $N_2O$, hinting that $NH_3$ is oxidized into $N_2O$. Starting from 325 °C (similar temperature range as in engine test bench), $NH_3$ concentration increases again, accompanied by a simultaneous decrease in $H_2$ conversion, which is linked to the $H_2$ production from the increasing SR reaction rate. The 50% conversion of CO and $H_2$ occurs at 156 °C and 170 °C, respectively (150 °C and 170 °C for Pd/Rh in engine test bench). The consistency of observations in lab-scale reactors and engine test benches demonstrates the reliability and significance of cold-start tests in engine, which can provide valuable implications on reaction pathways.

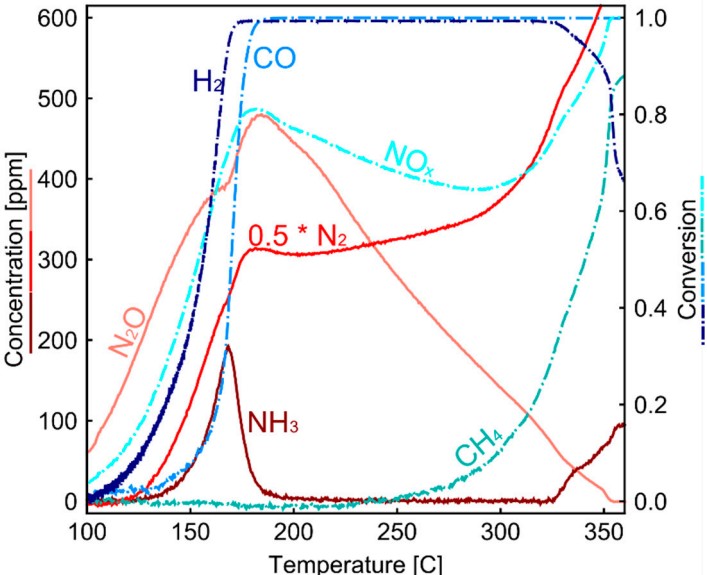

**Figure 9.** Temperature and Concentration profiles during temperature programmed reaction in model gas reactor.

### 3.5. Slow $\lambda$ Oscillation

The above $\lambda$ ramp experiments reveal the conversion characteristics in a wide range of $\lambda$, but it is limited in the analysis of transient operations due to the slow and continuous $\lambda$ change. Transient conversion characteristics are better captured during slow $\lambda$ oscillations, in which both the lean and rich phase last 60 s. Species concentration during one $\lambda$ oscillation at different temperatures are shown in Figure 10. The operating points of the engine correspond exactly to the low, mid and high load points in the previous $\lambda$ ramp experiments. The almost identical $O_2$ concentration profiles in the $\lambda$ transitions ($O_2$ concentration plots in Figure 10) indicate that both catalysts possess similar OSC. At low temperature (Figure 10a), after transition to rich conditions, the THC concentration in both

catalysts passed through a low value before stabilizing at ca. 1000 ppm, corresponding to a conversion of 0.4. The THC conversion change of Figure 10a can be imagined to correspond to a fast transition along the ramp-down curve in Figure 6b. The conversion remains high as long as OSC is not depleted. The temporary conversion enhancement is stronger in the Pt/Rh catalyst, corresponding to the higher conversion peak in Figure 6b (yellow line). Due to stronger conversion, the OSC depletes faster in the Pt/Rh catalyst, as more reducing agent is needed to oxidize CO and $H_2$ formed from SR. The fast depletion of OSC affects also the $NH_3$ formation. $NH_3$ is observed earlier in the Pt/Rh catalyst. After the transition to lean phase at t = 60 s, another temporary low THC concentration is observed. Similar to the previous analyses of the rich transition, this signals an SR enhancement related to the favorable λ condition during oxidation of the OSC component. In addition, CO is oxidized by the excess $O_2$ immediately after changing into the lean feed. This alleviates the competitive adsorption of CO and $CH_4$ and further enhances SR in the time range of 60 < t < 75 s when SR is occurring. After OSC is fully oxidized, SR is deactivated because oxygen vacancies are necessary for SR [26,50]. Without SR, THC conversion is very low. The converted part is mainly nonmethane hydrocarbons.

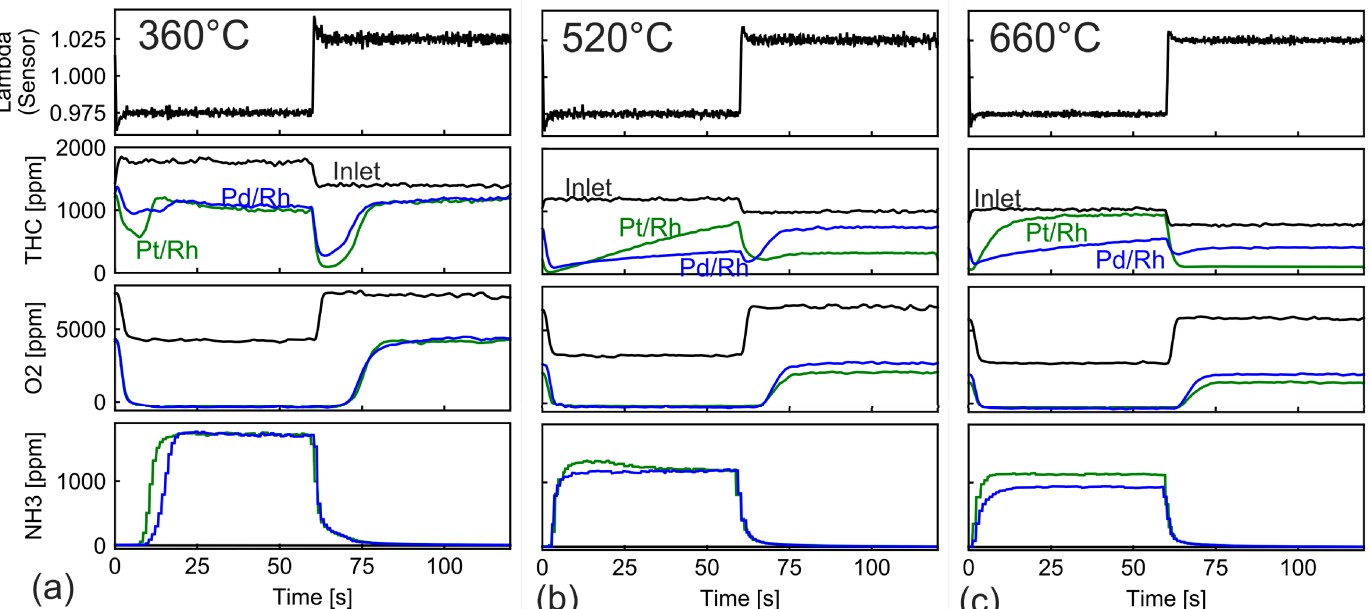

**Figure 10.** THC, $O_2$ and $NH_3$ Concentration profiles during slow λ oscillations at (**a**) 360 °C (**b**) 520 °C and (**c**) 630 °C in Pt/Rh and Pd/Rh catalyst.

At 520 °C (Figure 10b), the THC concentration in both catalysts reached minimum after the step change from lean to rich. This increase in conversion was attributed to SR activation in our previous work [28]. The SR conversion is stronger in the Pt/Rh catalyst immediately after the step change. The high THC conversion, however, attenuates with time. As discussed above, in Section 3.1, this is linked to the decrease of SR reaction rate caused by the formation and accumulation of ceria carbonates. The deactivation via carbonates affects only SR reaction because active ceria sites are involved in the $H_2O$ dissociation step during SR [22,25,26]. The attenuation is much faster in the Pt/Rh catalyst; the conversion drops from 0.98 to less than 0.3 in 60 s. The fast deactivation of SR explains the lack of SR hysteresis in Figure 2b for the Pt/Rh catalyst. In line with results from Figure 2, the Pt/Rh catalyst oxidizes $CH_4$ at significantly higher conversion than the Pd/Rh catalyst under lean conditions.

The SR activation and deactivation characteristics at 660 °C are very similar to those observed at 520 °C (Figure 10c). The Pt/Rh catalyst triggers initially higher SR rate but deactivates much faster than the Pd/Rh catalyst. The Pt/Rh catalyst also produces more $NH_3$ at 660 °C.

### 3.6. Conversion Characteristics of Pt/Pd/Rh Catalyst

Of the three metals, the role of Rh is unique and unreplaceable. Rh enhances $NO_x$ conversion and reduces selectivity toward unwanted $NH_3$ emissions. Pt and Pd both act as oxidation catalysts and each has different characteristics at different gas composition and temperature. Pt has clear advantages with rich feed at low temperature. However, SR deactivation is too fast at higher temperatures. Pd sustains SR for a longer time under rich conditions but in general has lower activity toward methane direct oxidation. The results suggest that the combination of these three metals can possibly integrate advantages and balance weaknesses.

A catalyst consisting of Pt/Pd/Rh (22/32.2/4.5 g/ft3) is prepared and installed in the engine test bench. The catalyst has a length of 100 mm. λ ramp experiment is performed on the Pt/Pd/Rh catalyst at 1600 rpm 210 Nm (520 °C), and the result is shown in Figure 11. During ramp-up (blue line), the $CH_4$ conversion decreases monotonously with λ under lean conditions, similar to Pt/Rh catalyst in Figure 2b. During ramp-down, at λ > 1.0, the $CH_4$ conversion increases with decreasing λ, which again corresponds to the behavior of Pt. The hysteresis in the range of 1.0 < λ < 1.1 can be traced back to Pd, which exists in different oxidation states during ramp-up and ramp-down. A clear SR hysteresis under rich conditions (0.95 < λ < 1.00) suggests that the Pt/Pd/Rh catalyst is able to sustain SR for a relatively long period of time, similar to the Pd/Rh catalyst in Figure 2a. The Pt/Pd/Rh catalyst combines the advantage of the Pt/Rh catalyst at lean conditions and the benefit of the Pd/Rh catalyst under rich conditions.

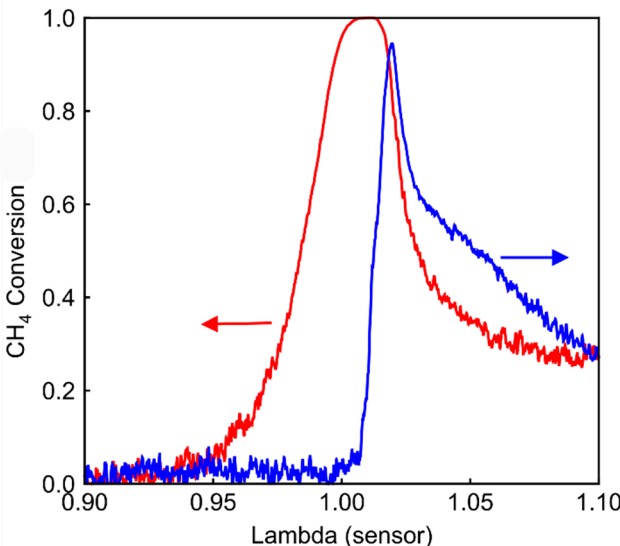

**Figure 11.** $CH_4$ conversion of the Pt/Pd/Rh catalyst during λ ramp-up (blue) and ramp-down (red) at 520 °C.

### 3.7. λ Oscillations in Pt/Pd/Rh Catalyst

3.7.1. Optimization of λ Oscillation

λ oscillations have been recognized to promote a significant enhancement of $CH_4$ conversion [17,27,29,30]. This was attributed to high SR reaction rate [29], which is enabled by the periodic catalyst regeneration from carbonaceous species. The potential of the Pt/Pd/Rh catalyst, which as demonstrated integrates the strength of both Pt and Pd, can be best demonstrated using λ oscillations. Figure 12 plots THC, $NO_x$ and $NH_3$ concentration profiles during oscillations of different period lengths (8, 20 and 26.6 s). The oscillation center ($\lambda_c$) is set at λ sensor signal 1.0, which is a practical reference for real-life applications. The actual λ ($\lambda_{corr}$) at sensor signal 1.0 was identified as 0.990 in a preliminary study. For better comparison, the oscillation period is normalized to 1.0. The concentration profiles are shown in one oscillation starting with rich phase (0 < $t_{norm}$ < 0.5), followed by with

lean phase ($0.5 < t_{norm} < 1.0$). For all three period lengths, the concentration profiles are similar. The highest concentration appears at $t_{norm} = 1.0$, while a local maximum can be observed at $t_{norm} = 0.5$. The 8 s oscillation has the highest THC average concentration (247 ppm, 77.3% conversion), while the 26.6 s oscillation has the lowest THC concentration (26 ppm, 97.6% conversion). At 26.6 s period length, the $NO_x$ concentration rises slightly at $t_{norm} = 1.0$, signaling that the OSC is almost completely filled. Further prolongation of period length leads to higher $NO_x$ peaks, which is highly undesirable in sight of strict $NO_x$ emissions regulations. The 26.6 s period length, which almost fully oxidizes OSC, is therefore the optimal setting for THC conversion. These results are consistent with our previous work [29]. The THC concentration profiles are shaped by four reaction stages including activation of SR, subsequent gradual deactivation of SR in rich phase, enhancement of SR and subsequent take-over of oxidation in lean phase. With a longer oscillation period, oxygen penetrates deeper into the catalyst, and a larger part of the catalyst undergoes regeneration, which results in a higher SR rate [29]. The average $NH_3$ concentration is respectively 377, 363, 357 ppm for the three period lengths. The change of oscillation period length does not significantly change the amount of $NH_3$ formation.

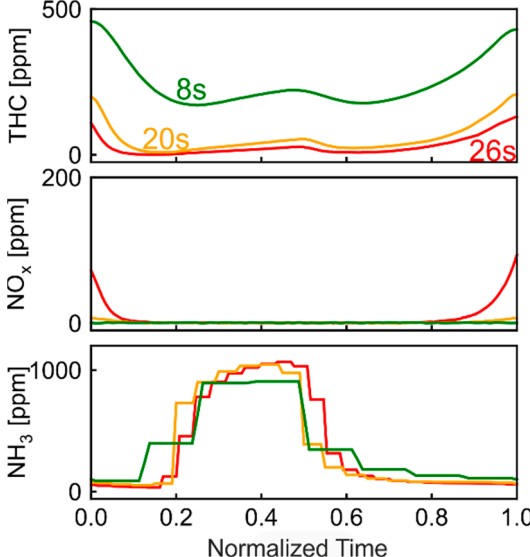

**Figure 12.** THC, $NO_x$ and $NH_3$ concentration profiles during oscillations with period length of 8 s, 20 s and 26 s.

As discussed before, the inaccuracy of $\lambda$ sensor leads to 1% deviation to the rich side from stoichiometry. To study the potential of Pt/Pd/Rh catalyst, better accuracy of the $\lambda$ sensor is assumed (0.5% and 0.2% deviation from stoichiometry). Figure 13 shows species concentration during oscillations of different center $\lambda_c$. The corrected $\lambda$ ($\lambda_{corr}$) equals 0.990 0.995 and 0.998, for the 1%, 0.5%, 0.2% assumption of $\lambda$ sensor deviation, respectively. The oscillation with $\lambda_c = 0.990$ ($\lambda_{sensor} = 1.0$), 13.3 period length, is already shown in Figure 13 and is plotted here again for better comparison. The oscillation period lengths are adjusted for the case of $\lambda_c = 0.995$ (10 s period length) and $\lambda_c = 0.998$ (8.7 s period length), in order to keep OSC from full oxidation. The $NO_x$ concentration in Figure 13c slightly increases to around 404 ppm at $t_{norm} = 1.0$ with $\lambda_c = 0.995$. However, the average $NO_x$ concentration stays low (46 ppm). At $\lambda_c = 0.995$ and 0.998, the $O_2$ concentration hardly rises from zero (Figure 13d), implying that the OSC is nearly but not completely filled. Based on previous discussions, these period lengths are optimal for THC conversion. The average THC concentrations for $\lambda_c = 0.995$ and 0.998 are 28 ppm and 29 ppm, respectively, thus similar to the case of $\lambda_c = 0.990$. The low $NH_3$ emissions are the main advantage of having $\lambda_c$ closer to 1.00. The average $NH_3$ concentration for $\lambda_c = 0.990, 0.995, 0.998$ is 357, 203, 129 ppm, respectively. At $\lambda_c = 0.998$, $NH_3$ is observed later under rich conditions, and its maximal

concentration is lower. Two processes affect the outlet $NH_3$ concentration profile: $NH_3$ formation and $NH_3$ oxidation. $NH_3$ is formed either through direct reduction of $NO_x$ by $H_2$ or through combined $NO_x$ reduction of CO and $H_2$ [38,39]. The $NH_3$ formation is therefore only possible with the presence of $NO_x$ under rich phase. Under same catalyst dimension, similar GHSV and temperature, our previous work [28] demonstrated that $NO_x$ is fully reduced in the first half of the catalyst under rich conditions. Thus, $NH_3$ formation occurs only in the upstream part of the catalyst (under rich conditions), where $NO_x$ is still available. Directly after shifting to rich conditions, the OSC component is almost completed in oxidized state. While the $NH_3$ is formed in the first half of the catalyst, the remaining OSC in the second half of the catalyst oxidizes the formed $NH_3$. Therefore, there is a delay for the observation of $NH_3$ concentration after shifting to rich conditions. The gradual increase of $NH_3$ concentration signals the slow depletion of OSC. With $\lambda_c = 0.998$, the OSC is depleted more slowly due to the higher $\lambda$ value, resulting in late $NH_3$ observation and therefore fewer $NH_3$ emissions.

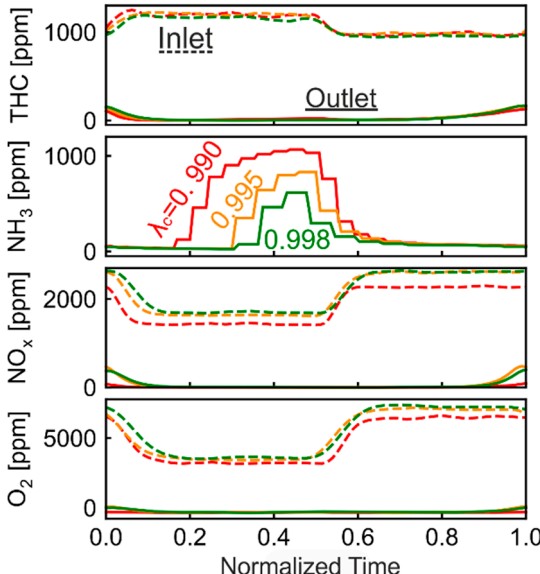

**Figure 13.** THC, $NH_3$, $NO_x$ and $O_2$ concentration profiles for Pt/Pd/Rh catalyst during oscillations with $\lambda_c = 0.990$, 0.995 and 0.998 (corrected $\lambda$).

3.7.2. Optimized Oscillation for the WHSC Cycle

To showcase the applicability of the above findings in various real-life driving conditions, the Pt/Pd/Rh catalyst was tested under the World Harmonized Steady Cycle (WHSC). For comparison, a Pd/Rh (65.4/4.6 g/ft3) catalyst of the same dimension was also tested. Representative engine operating points (except for idling point) are determined based on WHSC regulations [51] and the full engine map. Due to limitations of our engine test bench, the idling point is represented by an extremely low-load operating point (850 rpm, 50 Nm). For each load point, emissions are measured at steady and oscillating conditions. The steady-state measurements are conducted at $\lambda$ sensor signal equals 1.00, and they represent the state of art emissions without oscillation strategies. The oscillations are conducted at $\lambda_c = 0.998$ ($\lambda_{corr}$), which is calibrated for each operating point. The period length ($T$) is calculated according to the following equation:

$$T = T_0 \times \dot{m}_0/\dot{m}$$

where $T_0$ is the period length at reference point, $\dot{m}$ is the mass flow rate, and $\dot{m}_0$ is the mass flow rate at reference point. The $\lambda_c = 0.998$ oscillation in Figure 12 is selected as reference point, which has the best performance in both THC conversion and $NH_3$ emission. Figure 14 summarizes THC and $NH_3$ emission under steady and oscillating conditions at different

load points. In Figure 14a, except for the near-idling point (850 rpm, 50 Nm), the average THC emissions with oscillations are all below 35 ppm, corresponding to more than 0.97 conversion, while THC emission varies from 60 ppm to 1000 ppm under steady state. The high THC emission occurs especially at high load points (high temperature). This is in line with previously discussed λ ramp results in Figures 2 and 6a, where $CH_4$ conversion is close to zero after staying in rich condition for a long period of time (ramp-up). Figure 14b (Pd/Rh) reveals similar results, that for steady measurements, high THC emissions are observed at high load points. By applying optimized oscillations, the THC emissions are significantly lowered. However, the THC emissions with oscillations are at least 2.5 times higher than the Pt/Pd/Rh catalyst at each operating point (except for the near idling point).

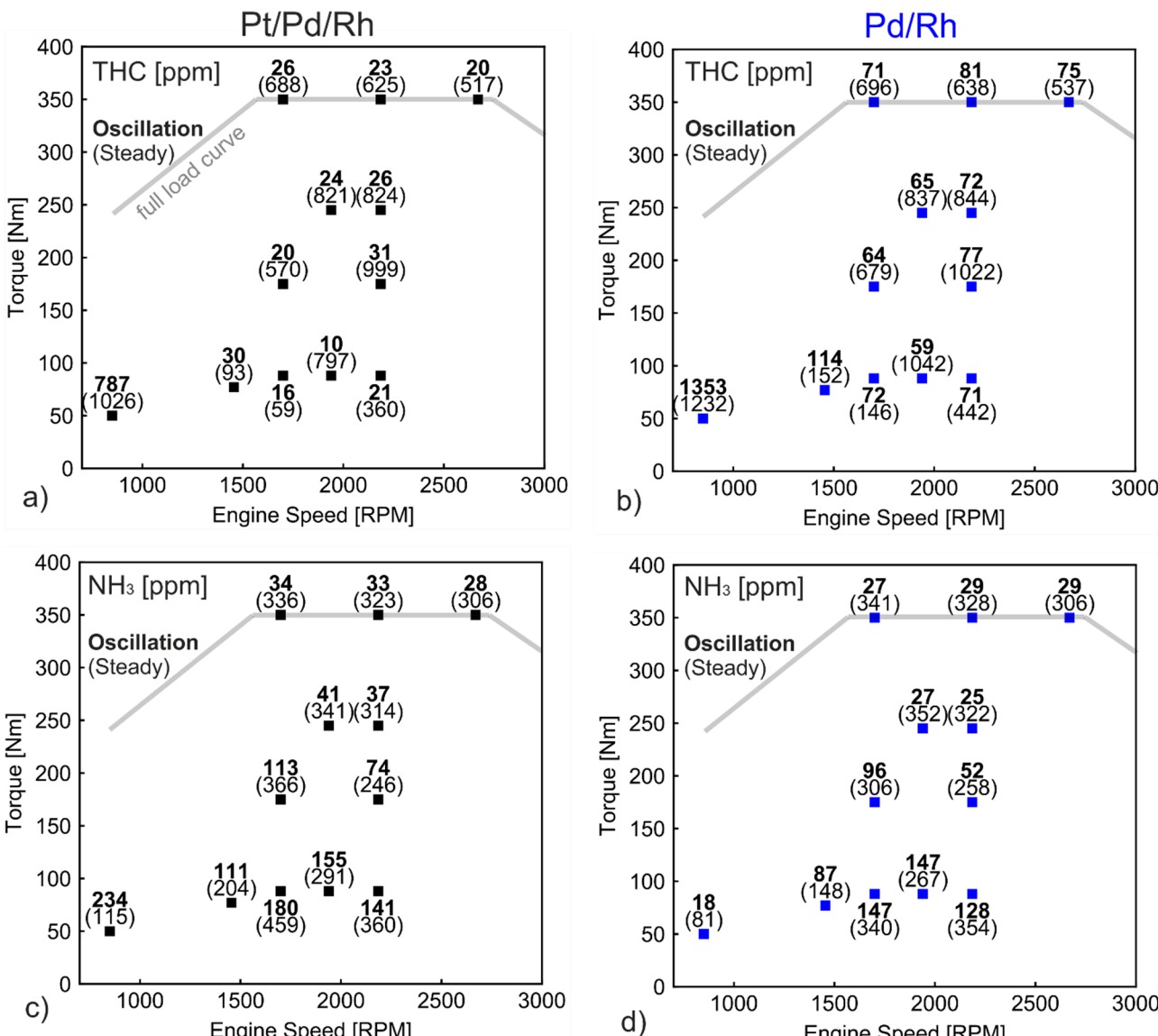

**Figure 14.** THC and $NH_3$ emission under steady and oscillations in WHSC operating points for (**a**) and (**c**) Pt/Pd/Rh catalyst; (**b,d**) Pd/Rh catalyst.

THC emissions (including near idling point) of both catalysts under oscillations are weighted according to the WHSC regulation [51]. The Pt/Pd/Rh catalyst leads to 199 mg/kWh THC, while Pd/Rh catalyst results in 339 mg/kWh. The comparison demonstrates the benefit of the Pt/Pd/Rh catalyst. The weighted THC emissions for steady state measurements are 1423 and 1585 mg/kWh, respectively. The results suggest that the cali-

brated oscillation strategies play a key role in lowering THC emissions. The optimization of catalyst composition helps to further reduce THC emissions under oscillation conditions.

The weighted $NO_x$ emission (not shown in Figure 14) for the Pt/Pd/Rh catalyst is 104 mg/kWh, while that of the Pd/Rh catalyst is 135 mg/kWh. Both values are well below the regulated emission of 400 mg/kWh in Euro 6 standard. The similarly low $NO_x$ emissions imply that the oscillation period length is well suited for both catalysts, without large breakthrough of $NO_x$. For both catalysts, optimized oscillation also results in lower $NH_3$ emission compared with steady state results (Figure 14c,d). This can be traced back to the precise $\lambda$ control during oscillation, allowing a large part of $NH_3$ to be oxidized by OSC.

## 4. Conclusions

In this work, different characteristics of Pt, Pd and Rh in natural gas engine exhaust are demonstrated through various targeted test procedures. Pt is found to be more active in methane direct oxidation (by oxygen) than Pd at mid to high temperatures (520–660 °C). SR is activated at lower temperatures in the Pt/Rh catalyst than the Pd/Rh. At mid to high temperatures, however, the SR reaction in the Pt/Rh catalyst only lasts shortly and is deactivated rapidly. The Pd/Rh catalyst keeps SR active for longer time after transition from lean to rich. Regarding $CH_4$ direct oxidation under lean conditions, the activity of Pd is limited by the less active metallic Pd(0), which is either only slowly oxidized at mid temperatures or stays unoxidized at high temperatures. The addition of Rh does not affect conversion of $CH_4$. However, Rh is crucial for high $NO_x$ conversion and selectivity toward $N_2$.

In view of the complementary functionalities of the different noble metals, a Pt/Pd/Rh catalyst is designed for testing on engine test bench. Steady high THC conversion can be achieved by means of $\lambda$ oscillations between lean and rich conditions. Under rich conditions, THC is effectively removed through SR, which occurs on highly active, regenerated catalyst surface. Under lean conditions, the accumulated carbonaceous species on the surface are removed by the excess oxygen. THC conversion increases with oscillation period length, until the period length is too long and results in fully filled/depleted OSC. The average $NH_3$ concentration is proved to depend on the oscillation center $\lambda$ ($\lambda_c$) but is not affected by the length of the oscillation period. $NH_3$ emissions decrease with increasing $\lambda_c$ toward 1.00. The optimized oscillation strategy is applied to various characteristic engine operating points, defined by WHSC. For most of the operating points, more than 97% of $CH_4$ conversion is reached under oscillatory conditions. The $NH_3$ emission is also reduced in comparison to steady state measurements.

The observation of different characteristics of Pt, Pd and Rh, as well as the proof of high conversion obtained with the Pt/Pd/Rh catalyst, holds important implications in future development of catalysts for methane abatement.

**Author Contributions:** Conceptualization and methodology, M.W. and P.D.E.; experiment, M.W., M.A.-A. and T.F.; writing, M.W.; review and editing, all authors contribute to review of the manuscript; visualization, M.W.; supervision, O.K.; project administration, P.D.E. and D.F.; funding acquisition, P.D.E. All authors have read and agreed to the published version of the manuscript.

**Funding:** This research was funded by the Bundesamt für Umwelt (Switzerland) and FPT Motorenforschung AG, grant number UTF 584.13.18.

**Acknowledgments:** R. Graf from our laboratory are gratefully acknowledged for support with the engine test bench measurements. G. Mancino from FPT INDUSTRIAL S.p.A and D. Klein from FPT Motorenforschung AG are also gratefully acknowledged for the project support.

**Conflicts of Interest:** The authors declare no conflict of interest.

## Appendix A

**Table A1.** List of Inlet Species Concentration at Different λ Values during λ-Ramp at 520 °C.

| λ | O$_2$ [ppm] | NO [ppm] | H$_2$ [ppm] | CO [ppm] | THC [ppm] | CO$_2$ [ppm] |
|---|---|---|---|---|---|---|
| 0.90 | 2518 | 696 | 18,543 | 23,741 | 1617 | 77,330 |
| 0.91 | 2506 | 785 | 16,997 | 21,958 | 1538 | 78,733 |
| 0.91 | 2581 | 897 | 15,552 | 20,305 | 1517 | 80,033 |
| 0.93 | 2575 | 994 | 14,074 | 18,636 | 1456 | 81,367 |
| 0.94 | 2572 | 1132 | 12,587 | 16,805 | 1387 | 82,827 |
| 0.95 | 2737 | 1228 | 11,341 | 15,444 | 1347 | 83,834 |
| 0.96 | 2815 | 1369 | 9939 | 13,687 | 1291 | 85,183 |
| 0.97 | 2964 | 1519 | 8610 | 12,032 | 1237 | 86,393 |
| 0.98 | 3295 | 1666 | 7409 | 10,625 | 1194 | 87,353 |
| 0.9 | 3669 | 1856 | 6163 | 9077 | 1159 | 88,405 |
| 1.00 | 4198 | 2067 | 4891 | 7428 | 1105 | 89,414 |
| 1.01 | 4989 | 2294 | 3713 | 5937 | 1058 | 90,181 |
| 1.02 | 6033 | 2510 | 2852 | 4816 | 995 | 90,510 |
| 1.03 | 7049 | 2737 | 2130 | 3764 | 970 | 90,833 |
| 1.04 | 8004 | 2853 | 1751 | 3178 | 935 | 90,888 |
| 1.05 | 9308 | 3019 | 1397 | 2689 | 921 | 90,671 |
| 1.06 | 10,308 | 3127 | 993 | 2108 | 889 | 90,678 |
| 1.07 | 11,481 | 3253 | 720 | 1613 | 883 | 90,465 |
| 1.08 | 13,141 | 3419 | 684 | 1548 | 885 | 89,782 |
| 1.09 | 14,751 | 3455 | 664 | 1568 | 878 | 89,004 |
| 1.10 | 16,248 | 3582 | 558 | 1320 | 881 | 88,409 |

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
