# Peer review of "Investigation on the Role of Pd, Pt, Rh in Methane Abatement for Heavy Duty Applications"

_catalysts, doi:10.3390/catal12040373_

Round 1
Reviewer 1 Report
The authors identified the characteristic behaviors of Pt, Pd and Rh at different temperatures using various procedures, and they concluded that the Pt/Pd/Rh catalyst holds important implications for future development of catalysts for methane abatement. The manuscript is well organized and therefore this paper can be accepted after addressing the following issues.
- The detailed information of Pd, Pt/Rh, Pd/Rh and Pt/Pd/Rh (such as morphology, size, and metal ratio) should be provided.
- The authors need to improve the English carefully, especially, the Figure Captions for Figure 1 and Figure 2.
Reviewer 2 Report
The present work reports a wide experimental investigation of different catalytic systems for the reduction of emissions of natural gas burn engines. Both bench scale units and lab-scale units are used to get a rationale understanding of the effect of temperature and oxygen excess in the abatement efficiencies of several pollutants. This is definetely a topic of great interest the journal readership.
The work is well written, scientifically sound and really interesting; it also provides the basis for the design of an optimized combustion control towards the reduction of pollutant emissions. This work, as other works in the scientific production of the authors, will set new standards in the design of these systems.
Comments to the authors :
L81 : can the authors, for the monoliths used in this study report both OFA of the washcoated support and an estimate of the catalyst thickness. Both these informations are fundamental to assess possible mass transport limitations. With respect to the reported results in the following sections, do the authors think that this is the case? So, in other words, any of these results are influenced by the monolith shape and catalyst thickness?
L127 Can the authors provide the definiton adopted for GHSV for sake of understanding of the results?
A general comment for the experiments performed with the setup described at Section 2.1. It would be interesting to have at least the range of variation of the inlet composition with the oxygen excess variation. The oscillations in the feed may help in understanding some of the observed phenomena
A general comment again for experiments performed at the bench test. Water is strongly affecting the kinetics of many of the reactions here investigated in detail. Do the authors think that water variations in real life operation may impact on the observed results and in particular on the optimized lambda variation?
L281 If i correctly uderstood results in this section are based on a simulated engine out ( i would change these words, seem confusing)
The authors claim that the steam reforming has a relevant effect in the methane abatement in some of the conditions, however this reaction is strongly subject to carbon deposition that rapidly hinders its effect. Are the authors planning to try to characterize the carbon formation and deposition kinetics on these systems? If I correctly understood, however, the inhibition is only present for the SR reaction and not for other reactions in the system, therefore there is a sort of preferential deposition in the system?
Figure 7 Why, if the temperature is controlled, the three systems display different values? Is is something related to the thermal mass of the monolith or due to exo-endothermicity of some of the reactions? Also, I would consider to increase the size of the image for readability
Reviewer 3 Report
Fine and interesting paper. Authors are specialists in this field, so paper can now be accepted. Minor faults and misprints can be corrected at proofsreading stage
Author Response
Dear Reviewer,
Thanks for your positive review. We have improved the language quality of the paper discussing several details with an English native person.